# EXPLORING REDUNDANCY AND SHARED REPRESENTATIONS FOR TRANSFORMER MODELS OPTIMIZATION

## ABSTRACT

Large Language Models (LLMs) deliver state-of-the-art performance but at the cost of extreme computational and energy demands, raising the question of how much of their capacity is truly necessary. This paper explores structural and weight redundancies in Transformer-based architectures, aiming to identify inefficiencies and leverage them through targeted compression techniques. A central focus is assessing whether different modules perform overlapping functions. Although some degree of similarity is observed in the analyzed cases, redundancy proves to be lower than expected, challenging the assumption that weight matrices can be interchanged across layers without compromising performance. Additionally, an analysis of model matrices examines whether they exhibit an inherently low-rank structure. To further explore these aspects, three novel compression methods are introduced: MASS, which enforces weight aggregation and sharing, along with two factorization-based techniques, GlobaL Fact and ABACO. Experimental results show that while these approaches achieve model compression, their ability to maintain performance is limited, reducing their practical viability. The findings highlight the complexity of extracting redundancy from Transformer architectures, raising questions about its extent across layers and blocks. By addressing these challenges, this paper aims to contribute to ongoing efforts to improve the efficiency of LLMs.

## 1 INTRODUCTION

Transformer-based Large Language Models (LLMs) have achieved remarkable performance across diverse natural language processing tasks, but their scale imposes severe computational and energy costs. This raises a central research question: to what extent is the capacity of such models necessary, and how much of it stems from redundancy in their architecture?

Several lines of work suggest that redundancy exists at different levels of LLMs, from individual weights to full Transformer blocks.He et al. (2024); Song et al. (2024); Dorszewski et al. (2025); Wang et al. (2025) Compression methods such as pruning, quantization, and knowledge distillation have been developed to mitigate this inefficiencySajjad et al. (2023); Yuan et al. (2023); Song et al. (2024); He et al. (2024); Wang et al. (2025), while low-rank factorization offers a more structured approach to reducing parameter count. However, the degree of functional overlap across layers, and its implications for compression, remains insufficiently understood.Dorszewski et al. (2025); He et al. (2024); Sajjad et al. (2023) In particular, simple weight-sharing strategies often fail to preserve accuracy, indicating that redundancy may be more subtle than anticipated.Song et al. (2024); Wang et al. (2025); Sajjad et al. (2023)

This paper contributes to addressing this gap through two directions. First, we conduct a systematic analysis of functional redundancy in Transformer architectures, assessing whether different layers and modules encode overlapping representations. Our findings indicate that redundancy is present, but to a lesser extent than commonly assumed, complicating straightforward layer-sharing strategies. Second, motivated by this analysis, we introduce three new compression techniques designed to exploit weight structure effectively: 1) *Matrix Aggregation and Sharing Strategy (MASS)*, which aggregates and shares weights across blocks; 2) *Global and Local Factorization (GlobaL Fact)*, which balances shared and layer-specific low-rank factors; and 3) *Adapter-Based Approximation*

*and Compression Optimization (ABACO)*, which progressively transfers knowledge into low-rank adapters.

We evaluate these methods on standard benchmarks, comparing them to established factorization-based compression approaches. While our results show only partial success – retaining accuracy remains a persistent challenge – they highlight promising directions for reducing redundancy without severely degrading model quality. Overall, our study clarifies the limitations of redundancy-based compression in Transformers and provides new techniques that can inform future work on efficient LLM deployment. All experiments were conducted on a server equipped with 64 GB of RAM and an Nvidia GeForce RTX 4090 GPU.

## 2 RELATED WORK

Model compression techniques aim to permanently reduce model size, enhancing efficiency while preserving accuracy. Common approaches related to this work include low-rank adaptation, pruning, and low-rank factorization.

**Low-Rank Adaptation.** Large Language Models (LLMs) exhibit strong generalization capabilities but often require fine-tuning to adapt to specific tasks. However, updating all parameters is resource-expensive. To mitigate this, adapter-based fine-tuning techniques have emerged, with Low-Rank Adaptation (LoRA) Hu et al. (2022) being a notable approach. LoRA reduces the number of trainable parameters by factorizing weight updates into low-rank matrices, enabling efficient adaptation while keeping the original model frozen. Given a weight matrix $\mathbf{W} \in \mathbb{R}^{m \times n}$, LoRA approximates the fine-tuning update as: $\mathbf{A} \in \mathbb{R}^{\mathbf{m} \times \mathbf{r}}$ and $\mathbf{B} \in \mathbb{R}^{\mathbf{r} \times \mathbf{n}}$, with rank $r \ll \min(m, n)$. However, while LoRA enhances efficiency in training, it does not reduce inference-time complexity, as the full model remains active during deployment.

**Pruning.** Pruning reduces redundancy by removing parameters to cut model size and computation. Structured methods include Deja Vu Liu et al. (2023), which dynamically prunes attention heads and MLP neurons via a lightweight predictor to exploit contextual sparsity, achieving notable speed-ups but at the cost of added complexity and hardware constraints, and SLEB Song et al. (2024), which prunes entire Transformer blocks by measuring their contribution to token predictions, yielding hardware-friendly efficiency without retraining. Early-exit strategies instead skip layers when intermediate outputs are sufficiently confident, reducing inference cost but requiring careful calibration and training. Unstructured pruning operates at the weight level: SparseGPT Frantar & Alistarh (2023) frames pruning as sparse regression, reaching high sparsity with minimal accuracy loss, while Wanda Sun et al. (2024) incorporates activation statistics to improve over simple magnitude-based pruning, achieving efficient, input-aware compression without retraining.

**Low-Rank Factorization.** Low-rank factorization decomposes large weight matrices into smaller components. Truncated SVD provides the best rank-$k$ approximation of a matrix Eckart & Young (1936) and has been widely applied to compress LLM weight matrices such as attention and feed-forward layers Noach & Goldberg (2020); Sharma et al. (2024). While simple and effective, it is computationally expensive for very large matrices. To address SVD's high reconstruction error on Transformer weights, DRONE Chen et al. (2021) incorporates input activation statistics, minimizing output rather than weight reconstruction error; however, its performance depends on the representativeness of the calibration data. Fisher-Weighted SVD (FWSVD) Hsu et al. (2022) further integrates Fisher information to prioritize task-critical parameters, achieving better task fidelity but at the cost of gradient-based overhead. DSFormer Chand et al. (2023) replaces pure low-rank approximations with a dense–sparse factorization optimized jointly with task objectives, improving compression–accuracy trade-offs though requiring more complex implementation. Activation-aware SVD (ASVD) Yuan et al. (2023) adapts decompositions to activation distributions and layer sensitivity, offering training-free compression and memory savings, but with added calibration complexity. Finally, SVD-LLM Wang et al. (2024) combines truncation-aware whitening with closed-form layer-wise updates, enabling selective truncation and alignment of compressed weights, achieving high compression with minimal performance loss.

## 3 INVESTIGATING REDUNDANCY

This section presents the motivations driving the redundancy analysis and shows the results of substituting components across the layers of the network.

### 3.1 MOTIVATION

A central aspect of redundancy investigated in this work is functional redundancy, namely cases where multiple layers or sub-blocks perform highly similar or nearly identical transformations. Conceptually, given two sections of a Transformer, the first computing a function $f$ and the second $f'$, the question is whether $f \approx f'$, and if so, how this similarity can be measured. In principle, the most rigorous way to assess such similarity would be through a direct analytical comparison of the transformations. While feasible for simple or linear mappings, this approach quickly becomes intractable for the non-linear and high-dimensional composite functions implemented by self-attention or feed-forward layers. Random input probing is equally uninformative, since it fails to reflect the structured distribution of natural language inputs. An exception arises for single linear transformations, where comparing weight matrices directly provides a reliable measure of similarity without requiring actual activations. For more complex modules, however, defining a robust similarity metric is challenging.

Several approaches can be considered. Directly comparing parameter sets (e.g., computing the norm of the difference between two weight matrices) is computationally cheap and interpretable, but it ignores non-linearities and compositional structure. Comparing activations offers a more behaviorally grounded perspective: cosine similarity between outputs highlights alignment in directional structure, while norms of activation differences emphasize absolute divergence. Both methods, however, suffer from sensitivity to the high dimensionality of embeddings, dependence on calibration set size, and difficulty in capturing scale. A more practical and task-relevant approach is to evaluate global performance after module replacement: swapping one layer or sub-block with another and measuring the resulting model's benchmark performance. If the modified model retains its performance, this suggests that the replaced components are functionally redundant. This method has the advantage of grounding similarity in real operational behavior, but it conflates local and global effects, as any observed difference reflects interactions across the entire model.

After experimentation with these options, the analysis in this work primarily adopts the replacement-based approach. The methodology proceeds by systematically exchanging modules between different Transformer blocks and evaluating the impact on model performance. Specifically, for a model with $B$ blocks, modules from block $i$ are replaced with those from block $i + k$ (for $k \in [-i, -i+1, \ldots, B-i]$), enabling pairwise comparisons across the stack. Performance is then measured on established language modeling benchmarks, allowing us to assess whether the exchanged modules execute redundant computations or provide unique contributions. Stability under replacement indicates redundancy, whereas substantial degradation suggests functional specialization.

We focus our analysis and experiments on the feed-forward networks (FFNNs) within each block since they contain a large number of parameters and, as pointed out later in section 3.2, dealing with FFNN is much less computational complex than dealing with attention blocks. Formally, for input $\mathbf{x}$, the FFNN in block $i$ is expressed as

$$\mathbf{FFNN}_i = \left(f_a\left(\mathbf{x} \cdot \mathbf{W}_{\text{gate},\, i}^T\right) \circ \left(\mathbf{x} \cdot \mathbf{W}_{\text{up},\, i}^T\right)\right) \cdot \mathbf{W}_{\text{down},\, i}^T \tag{1}$$

and the analysis investigates whether transformations across layers satisfy approximate equivalence:

$$\mathbf{FFNN}_i \approx \mathbf{FFNN}_j, \quad i \neq j \tag{2}$$

Evaluating this equivalence through replacement-based testing provides a concrete, performance-driven measure of functional redundancy. Moreover, it offers insight into optimization opportunities such as pruning or merging redundant components while retaining model quality.

### 3.2 INVESTIGATION

Prior to experimenting with compression, it is crucial to understand the degree of functional redundancy in Transformer models. This is assessed by substituting selected modules with those from different blocks and measuring the resulting performance impact.

The study primarily focuses on the functional equivalence between FFNN modules in Llama 3.1 (8B), with performance variations analyzed on two benchmarks: HellaSwag and GSM8K. Pairs of feed forward functions from different blocks are compared to assess whether their functionality is approximately preserved across the Transformer stack. If the model maintains good performance after replacing the FFNN component of block $i$ with that of block $j$, the two modules may be functionally similar. To better assess the impact of the replacement, the performance of the modified model is compared to that of the LLM in which the overwritten module is instead ablated.

Figure 1 shows the accuracy difference on HellaSwag between models that underwent replacement of FFNNs and those with the same modules ablated, while Figure 2 illustrates the same comparison evaluated on GSM8K. In the heatmaps, each cell represents a substitution where the module specified by the column is overwritten by the one indicated by the row. Red cells highlight cases where replacement preserved performance better than ablation, while blue cells denote instances where replacement caused greater degradation, suggesting that the components perform unique functionalities. The dominance of blue cells indicates that most FFNNs are not interchangeable. Although some redundancy is observed no consistent pattern emerges. Even among adjacent blocks, no robust similarity is observed.

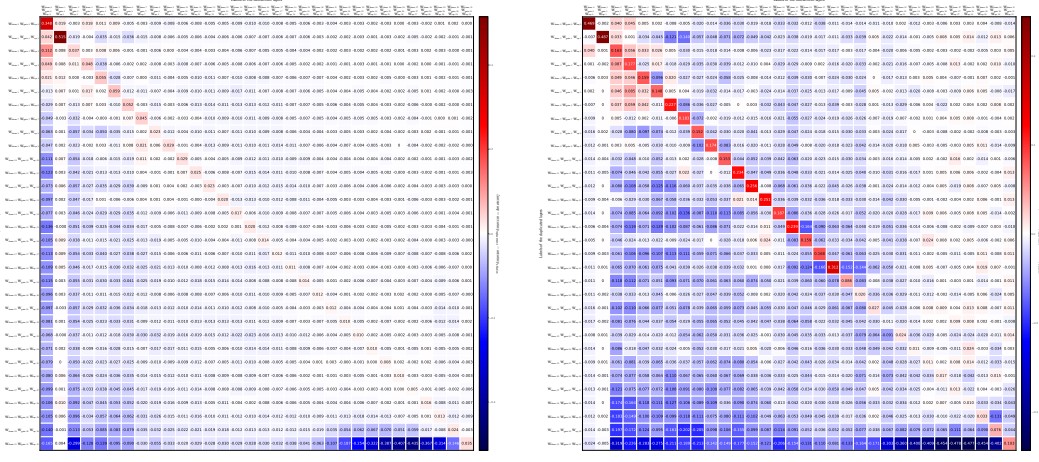

Figure 1: Performance difference on HellaSwag between replacing an FFNN layer with a different layer and removing the layer for Llama 3.1. Red indicates improvement, blue degradation.

Figure 2: Performance difference on GSM8K between replacing an FFNN layer with a different layer and removing the layer for Llama 3.1. Red indicates improvement, blue degradation.

These results suggest that FFNN modules across Transformer blocks perform quite distinct functions, making direct replacement an ineffective compression strategy. The same investigation was extended to Gemma 2 (2B), obtaining similar results discussed in Section A.1. Perhaps the most interesting finding from the visualization (apart from the conclusion that layers are not directly replaceable) is the fact that the behavior of the first two layers and last layer of the transformer appear somewhat distinct from the other layers. It appears that on the HellaSwag dataset reusing the first FFNN layer at subsequent layers of the network has a net positive effect compared to layer ablation for the first four subsequent layers of the network, indicating that there may be some redundancy between these layers, and then has a catastrophic negative effect for subsequent layers, indicating no redundancy whatsoever. A similar, if much less pronounced effect is seen on the GSM8K dataset. Meanwhile, reusing the second FFNN layer across the subsequent layers of the netork appears to have minimal effect (neither net positive nor net negative) with respect to layer ablation (which is not the case for the third FFNN layer onwards), indicating the behaviour of the second FFNN layer is particular in some way. Finally catastrophic performance degradation is observed for both datasets whenever the last FFNN layer is replaced by a previous layer, indicating that its function is very much associated with its layer position.

Due to the high computational cost of benchmarking each modified model, only partial results (where nearby layers are replaced) are reported for the self-attention components of Llama 3.1. On

HellaSwag, replacing self-attention layers with those from other blocks yields performance similar to ablation, with slight improvements in the middle layers but significant degradation in the earliest ones, as we show in Figure 3. This indicates that the initial self-attention modules perform highly specialized functions that cannot be replicated by later layers. In Figure 4 we show that the pattern is similar for GSM8K: replacements in deeper layers are often less disruptive than removal, but the first layers again prove critical. Overall, these results suggest that while limited redundancy may exist among self-attention modules in central and deeper parts of the network, the lack of robust patterns and the sensitivity of the earliest layers make them poor candidates for systematic replacement.

We note a pronounced red diagonal in the top left of the heatmap for GSM8K. The colour intensity along the diagonal indicates the cost of ablating the self-attention layer and therefore quantifies its overall importance in the network. Thus the self-attention modules appear to have little importance past the second layer of the network for HellaSwag, which is understandable given it's Language Modeling task and Perplexity-based evaluation. Meanwhile for GSM8K, which contains maths problems, the self-attention layers appear to be important up to mid-point of network, after which their importance drops to near zero. This drop off is interesting from a compression standpoint.

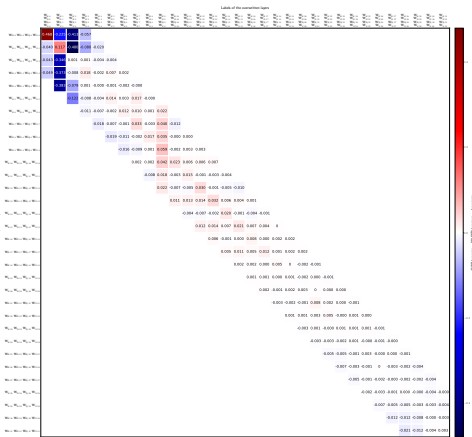 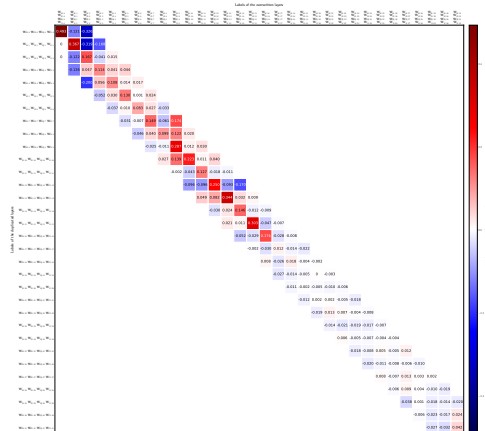

Figure 3: Performance difference on HellaSwag between replacing a self-attention module with a different layer and removing it for Llama 3.1. Red indicates improvement, blue degradation.

Figure 4: Performance difference on GSM8K between replacing a self-attention module with a different layer and removing it for Llama 3.1. Red indicates improvement, blue degradation.

## 4 COMPRESSION STRATEGIES

This section presents the compression strategies designed to address redundancy.

### 4.1 MATRIX AGGREGATION AND SHARING STRATEGY (MASS).

The proposed MASS leverages redundancies across linear layers to compress models by aggregating multiple components into shared representations. The goal is to reduce parameters while preserving functionality. The method proceeds in two phases:

1. **Grouping** Layers selected for compression are partitioned into subsets of structurally compatible matrices. A key requirement is that grouped matrices share the same shape to enable aggregation. In this work, groups are defined by matrix type (e.g., query, key, value, feed-forward). The first and last layers are excluded, as their roles differ significantly (see Section 3.2). Exploring more sophisticated grouping criteria is left for future work.

2. **Aggregation** Within each group, weight matrices and biases are merged into a shared layer according to an aggregation criterion. In this study, we employ simple averaging: for group

$\mathbf{S}i = (\mathbf{W}1, \mathbf{b}1), \dots, (\mathbf{W}N_i, \mathbf{b}N_i)$, the shared parameters are $\mathbf{W}\mathbf{S}i = \frac{1}{N_i}\sum j = 1^{N_i}\mathbf{W}j$, $\mathbf{b}\mathbf{S}i = \frac{1}{N_i}\sum j = 1^{N_i}\mathbf{b}_j$.

This strategy assumes structural alignment across matrices, so that corresponding elements carry comparable semantic meaning. Misalignment would risk incoherent representations. Alternative strategies (e.g., weighted averaging or clustering-based aggregation) may better capture shared features and are left for future research: as with the grouping criteria, our goal was to explore a simple approach.

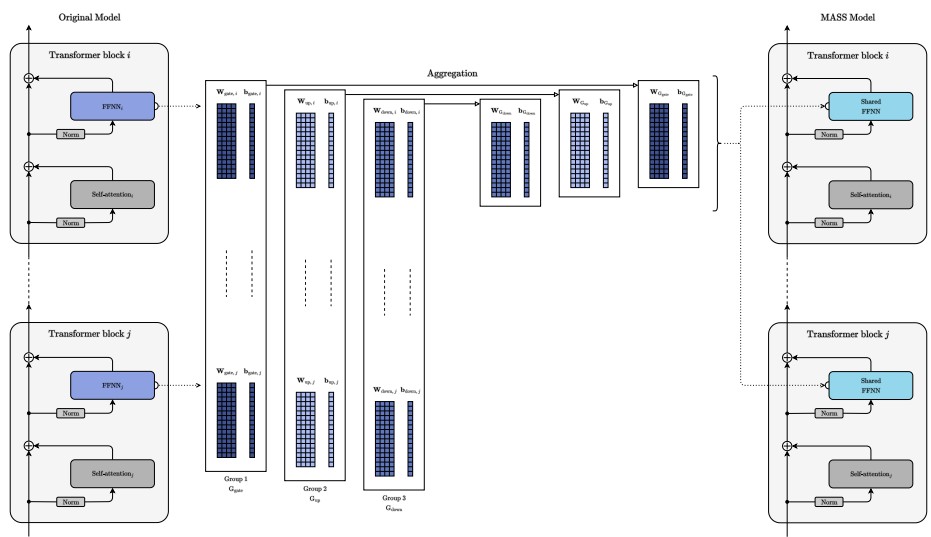

Figure 5: Illustration of MASS applied to feed-forward layers, grouped by matrix type.

Finally, replacing multiple layers with a shared representation inevitably reduces layer-specific expressivity. To mitigate this, we fine-tune the compressed model on downstream or pretraining data, allowing it to adapt to the aggregation and recover performance.

## 4.2 GLOBAL AND LOCAL FACTORIZATION (GLOBAL FACT).

GlobaL Fact compresses Transformer models by factorizing selected weight matrices into the product of a global matrix, shared across a group of layers, and a local matrix that retains layer-specific features. Layers are grouped based on their functional roles. Given a group $\mathbf{S}_i$ of $N$ layers, their weight matrices $\mathbf{W}_j \in \mathbb{R}^{m \times n}$ are approximated as:

$$\tilde{\mathbf{W}}_j = \mathbf{L}_j \cdot \mathbf{G}_{\mathbf{S}_i}, \quad j = 1, 2, \dots, N. \tag{3}$$

Here, the global matrix $\mathbf{G}_{\mathbf{S}_i} \in \mathbb{R}^{r \times n}$ captures shared patterns, while local matrices $\mathbf{L}_j \in \mathbb{R}^{m \times r}$ introduce layer-specific variations. The internal dimension $r$ (with $r < \min(m, n)$), controls the compression level. The initialization of $\mathbf{G}_{\mathbf{S}_i}$ is derived from a combination of the most relevant singular components of the weight matrices $\mathbf{W}_j$ within the group $\mathbf{S}_i$, extracted via SVD. Local matrices are computed by projecting the original weights onto the pseudo-inverse of the global matrix:

$$\mathbf{L}_j = \mathbf{W}_j \cdot \text{pseudo-inverse}(\mathbf{G}_{\mathbf{S}_i}), \quad j = 1, 2, \dots, N. \tag{4}$$

Finally, a fine-tuning phase refines the model post-factorization.

## 4.3 ADAPTER-BASED APPROXIMATION AND COMPRESSION OPTIMIZATION (ABACO).

ABACO is an iterative compression method that progressively transfers knowledge from full-rank weight matrices to low-rank adapters during fine-tuning, preventing abrupt parameter truncation. This is accomplished by integrating LoRA adapters in the model while gradually diminishing the contribution of the original weights through an adaptive penalization mechanism.

When adapters are introduced for compressing the weight matrix $\mathbf{W}_0 \in \mathbb{R}^{m \times n}$, its updated expression is:

$$\mathbf{W} = \alpha \mathbf{W}_0 + \mathbf{A} \cdot \mathbf{B}, \;\; \alpha = \alpha_0 \cdot b^{-c \cdot t} \tag{5}$$

where $\mathbf{A} \in \mathbb{R}^{m \times r}$ and $\mathbf{B} \in \mathbb{R}^{r \times n}$ are trainable low-rank matrices, and $\alpha$ controls the contribution of pretrained weights. During training, $\alpha$ follows an exponential decay schedule, progressively reducing the influence of the original parameters, where $\alpha_0$ is the initial value of $\alpha$, $t$ denotes the training step and the positive constant $c$ controls the rate of decay. In the experiments $b$ was set to 2.

At later stages, the original weights are either discarded or approximated.

## 5 EXPERIMENTS

This section presents the results of the evaluation of the compression strategies presented in Section 4, based on the Hellaswag and GSM8K benchmarks.

### 5.1 MATRIX AGGREGATION AND SHARING STRATEGY (MASS)

Experiments on MASS are conducted using Llama 3.1 (8B), where the model is first compressed using MASS, evaluated, fine-tuned on OpenWebText, and then re-evaluated on the benchmarks. Table 1 presents the benchmark results for MASS applied to specific layer types across all Transformer blocks. We note that the extreme aggregation strategy severely degrades performance, causing the model to perform as a random classifier. Even with fine-tuning, recovery remains impossible, indicating that the shared representations deviate too far from a useful parameterization. This failure is likely due to the lack of differentiation between layers and the simplistic averaging method, which fails to preserve meaningful structural properties.

| Model | Targets | #Params$_{\text{(CR)}}$ | HS$i$ | HS$f$ | GSM$i$ | GSM$f$ |
|---|---|---|---|---|---|---|
| Original Model | - | 8.030B | 0.789 | | 0.496 | |
| MASS | gate | 6.210B$_{(77.33\%)}$ | 0.266 | 0.259 | 0 | 0 |
| MASS | query | 7.510B$_{(93.52\%)}$ | 0.255 | 0.259 | 0 | 0 |

Table 1: Performance of Llama 3.1 (8B) compressed using full-scale MASS on the benchmarks HellaSwag (HS) and GSM8K (GSM) before and after fine-tuning.

To prevent performance collapse, we repeated the experiments with MASS applied only to the middle layers (9th to 24th), while keeping the first and last layers unchanged. Table 2 demonstrates that this adjustment enhances performance compared to full-scale aggregation. We note that while HellaSwag shows significant recovery, GSM8K remains severely degraded, confirming the crucial role of deeper layers in reasoning tasks. Fine-tuning improves the shared layers, particularly for query compression, but fails to fully restore the original quality of the model.

| Model | Targets | #Params$_{\text{(CR)}}$ | HS$i$ | HS$f$ | GSM$i$ | GSM$f$ |
|---|---|---|---|---|---|---|
| Original Model | - | 8.030B | 0.789 | | 0.496 | |
| MASS | gate | 7.149B$_{(89.03\%)}$ | 0.283 | 0.503 | 0 | 0.001 |
| MASS | query | 7.779B$_{(96.87\%)}$ | 0.318 | 0.701 | 0 | 0.010 |

Table 2: Performance of Llama 3.1 (8B) compressed using MASS applied to a subset of selected layers on the benchmarks HellaSwag (HS) and GSM8K (GSM) before and after fine-tuning.

### 5.2 GLOBAL AND LOCAL FACTORIZATION (GLOBAL FACT)

Experiments on GlobaL Fact are conducted using Llama 3.1 (8B), where the model is first factorized, evaluated, fine-tuned on OpenWebText, and then re-assessed. Table 3 presents the results

on the benchmarks. We note that the initial performance after factorization is severely degraded across all settings, causing the model to behave like a random guesser. Neither of the tested global matrix initialization strategies provided a stable starting point, suggesting that abrupt compression distorts representations too drastically. However, this degradation is not unique to GlobaL Fact, as a truncated SVD approach exhibits similar behavior.

| Model | Ranks | #Params$_{(CR)}$ | HS$i$ | HS$f$ | GSM$i$ | GSM$_f$ |
|---|---|---|---|---|---|---|
| Original Model | - | 8.030B | 0.789 | | 0.496 | |
| Truncated-SVD | $402_{gate}$ | 6.388B$_{(79.55\%)}$ | 0.262 | 0.401 | 0 | 0.006 |
| GlobaL Fact$_1$ | $512_{gate}$ | 6.388B$_{(79.55\%)}$ | 0.257 | 0.376 | 0 | 0 |
| GlobaL Fact$_2$ | $512_{gate}$ | 6.388B$_{(79.55\%)}$ | 0.249 | 0.426 | 0 | 0.017 |
| Truncated-SVD | $402_{up}$ | 6.388B$_{(79.55\%)}$ | 0.257 | 0.391 | 0 | 0.010 |
| GlobaL Fact$_1$ | $512_{up}$ | 6.388B$_{(79.55\%)}$ | 0.267 | 0.328 | 0 | 0.002 |
| GlobaL Fact$_2$ | $512_{up}$ | 6.388B$_{(79.55\%)}$ | 0.265 | 0.337 | 0 | 0.002 |
| Truncated-SVD | $402_{gate}$ $402_{up}$ | 4.746B$_{(59.11\%)}$ | 0.265 | 0.251 | 0 | 0 |
| GlobaL Fact$_1$ | $512_{gate}$ $512_{up}$ | 4.746B$_{(59.10\%)}$ | 0.265 | 0.248 | 0 | 0 |
| GlobaL Fact$_2$ | $512_{gate}$ $512_{up}$ | 4.746B$_{(59.10\%)}$ | 0.267 | 0.249 | 0 | 0 |
| GlobaL Fact$_2^*$ | $\frac{512_{gate}}{512_{up}}$ | 4.746B$_{(59.10\%)}$ | 0.267 | 0.255 | 0 | 0 |

Table 3: Performance of Llama 3.1 (8B) compressed using GlobaL Fact on the benchmarks HellaSwag (HS) and GSM8K (GSM) before and after fine-tuning.

Fine-tuning enables partial recovery when compressing a single layer type, particularly for gate projection matrices, suggesting that the model remains within a region of the parameter space that retains meaningful representations. However, performance remains well below the original model, suggesting that factorization introduces a substantial error. Compression of up projection layers proves even less effective, likely due to lower structural redundancy across these layers. When both gate and up projections are factorized simultaneously, fine-tuning fails to restore performance, indicating that excessive compression causes irreversible degradation. A sequential approach, as used in the case of Global Fact$_2^*$, compressing the gate projection first and then the up projection, yields no improvement, further highlighting the challenges of preserving knowledge under aggressive factorization.

## 5.3 ADAPTER-BASED APPROXIMATION & COMPRESSION OPTIMIZATION (ABACO)

Experiments on ABACO are conducted using Gemma 2 (2B), where the model is fine-tuned on OpenWebText using ABACO strategy, and then evaluated. Table 4 presents the results for ABACO applied to gate and query projection matrices.

| Model | Ranks | #Params$_{(CR)}$ | HS$i$ | HS$f$ | GSM$i$ | GSM$_f$ |
|---|---|---|---|---|---|---|
| Original Model | - | 2.614B | 0.730 | | 0.246 | |
| ABACO (gate) | $288_{gate}$ | 2.149B$_{(82.18\%)}$ | 0.730 | 0.337 | 0.246 | 0.008 |
| ABACO (query) | $144_{query}$ | 2.494B$_{(95.38\%)}$ | 0.730 | 0.306 | 0.246 | 0.003 |

Table 4: Performance of Gemma 2 (2B) when ABACO is applied to gate and query projection matrices on the benchmarks HellaSwag (HS) and GSM8K (GSM) before and after fine-tuning.

Experimental results show that ABACO is able to preserve a fraction of the original model's capabilities after compression, but the performance gap with the uncompressed LLM remains substantial. The method succeeds in enabling partial knowledge transfer, as evidenced by non-random final performance, yet the degradation is severe—particularly on challenging reasoning tasks such as GSM8K, where accuracy drops sharply. This highlights a core limitation of ABACO: while the approach avoids catastrophic collapse, it struggles to maintain the depth of reasoning required for complex benchmarks.

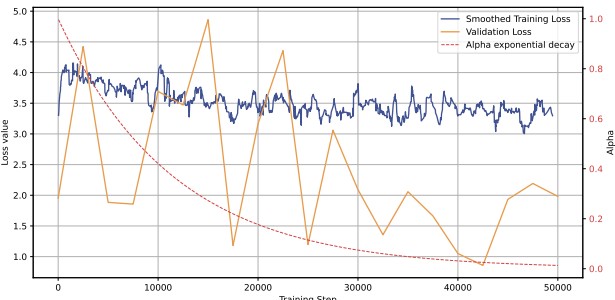

Figure 6: Training and validation loss during fine-tuning of Gemma 2 (2B) with ABACO applied to gate projections. The figure also depicts the exponential alpha decay. The training loss is smoothed using a window of 10 elements. The total fine-tuning time is 12 hours and 6 minutes.

The training dynamics illustrated in Figure 6 provide additional insight. At the start of fine-tuning, the training loss rises sharply (from roughly 2 in the original model to about 4), reflecting the reliance on randomly initialized adapters that initially fail to capture the functionality of the discarded weights. As training progresses and $\alpha$ decays, the adapters absorb more of the representational burden, leading to a gradual decrease in training loss. This trend suggests that ABACO achieves some degree of smooth knowledge transfer, enabling the model to stabilize in its compressed form. Importantly, the controlled decay of $\alpha$ prevents the abrupt disruptions typically caused by one-shot parameter truncation.

## 6 CONCLUSION

This work investigates redundancy and inefficiency in Transformer-based models to identify methods that can reduce the computational burden of LLMs. The analysis of functional redundancy across Transformer blocks reveals that modules, such as FFNN and self-attention, are inherently distinct and not easily interchangeable between different blocks, challenging the feasibility of direct compression through simple layer sharing.

Three novel compression methods are proposed, with each demonstrating only limited effectiveness in preserving the original performance. MASS leads to catastrophic performance loss when applied full-scale to a specific layer type and achieves only partial recovery when restricted to a subset of those layers. GlobaL Fact retains some integrity after fine-tuning, suggesting that a low-rank factorization framework with shared matrices could be a promising direction. Although it matches the performance of truncated SVD, it still falls short of the model's original accuracy. ABACO maintains a functional model but, like the other methods, suffers significant accuracy degradation, particularly in complex reasoning tasks.

These findings suggest that Transformer models may exhibit less repeated computation than previously assumed and that shared features are not trivially exploitable for compression. Even if this work cautions against overestimating redundancy, we suggest that research must keep moving towards more efficient and sustainable LLMs.

## 7 REPRODUCIBILITY STATEMENT

We provide detailed descriptions of each algorithm in Section 4 and infrastructures used for our experiments. The code for the experiments described in Section 5 is available in the following repositories:

- https://anonymous.4open.science/r/Experiment-Orchestrator
- https://anonymous.4open.science/r/Redundancy-Hunter
- https://anonymous.4open.science/r/Alternative-Model-Architectures-A16C/

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

# A  APPENDIX

## A.1  GEMMA-2-2B

This section investigates the functional redundancy of FFNN and self-attention modules in Gemma 2 (2B) through the replacement analysis. Following the methodology outlined in Section **??**, the results provide insights into the extent to which different layers can be interchanged without significantly impacting model performance.

### A.1.1  CHARACTERIZATION OF THE REDUNDANCY IN FFNN MODULES

This section presents the results of the replacement analysis applied to the FFNN modules of Gemma 2 (2B).

Figure 9 highlights the performance difference on HellaSwag between Gemma 2 (2B) with FFNN layers replaced by those from another block and the model with these layers entirely removed. The heatmap reveals a pronounced negative trend, with most values being negative, apart from a few exceptions. This suggests that substituting FFNN modules with those from a different block significantly degrades model performance, more than removing them. Specifically, in cases where ablation

already impacts the model, replacement further exacerbates the degradation, in scenarios where instead ablation has a minor effect, substitution with layers from another block degrade instead the model.

A similar pattern is observed in Figure 12, where widespread negative values with high absolute magnitudes indicate that replacing FFNN is even more detrimental on GSM8K compared to HellaSwag.

Despite the overall negative impact given by the substitution of FFNN, a few exceptions emerge. Specifically, some replacements of the FFNN module in the ninth block (index 8) results in better performance than ablation across both datasets. In the case of GSM8K, where the last layers also partially contribute to solving the task, modules in blocks indexed from 22 to 24 can be substituted with those from certain earlier blocks, leading to improved performance compared to their ablation. This suggests a minor degree of redundancy in the model. However, given the limited improvement relative to module removal and the rarity of such cases, it is insufficient to conclude that redundancy is a general property of FFNN modules in Gemma 2 (2B).

Ultimately, the results of this replacement analysis indicate that using the transformation performed by a FFNN component from one block to approximate another is not effective, implying minimal inter-sub-block redundancy.

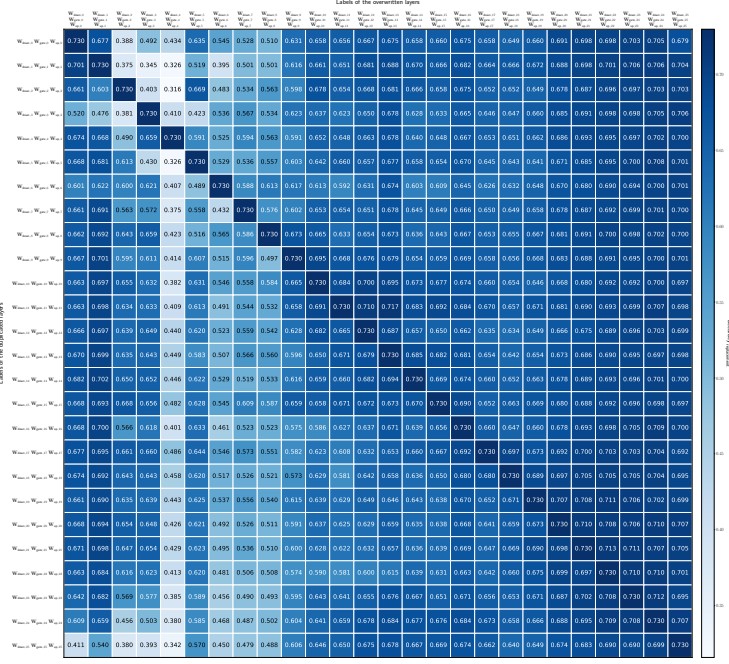

Figure 7: Heatmap illustrating the accuracy on HellaSwag of Gemma 2 (2B) having linear layers of the FFNN module replaced by the ones of a different block.

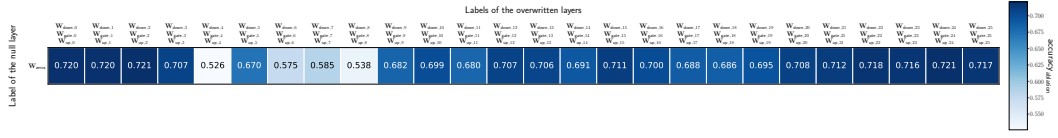

Figure 8: Heatmap illustrating the accuracy on HellaSwag of Gemma 2 (2B) having linear layers of the FFNN module removed.

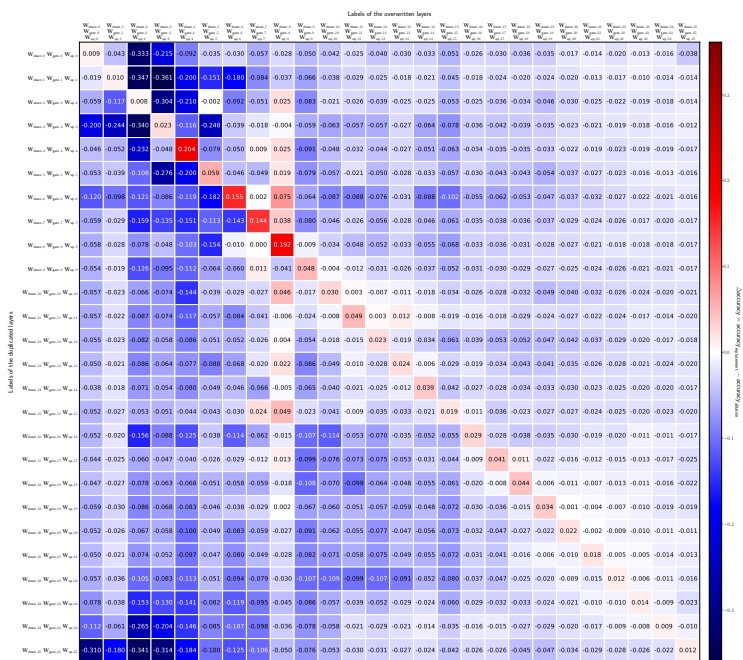

Figure 9: Heatmap illustrating the difference in performance evaluated on HellaSwag between Gemma 2 (2B) with linear layers in the FFNN module replaced by those from another block and Gemma 2 with the same layers removed.

### A.1.2 CHARACTERIZATION OF THE REDUNDANCY IN SELF-ATTENTION MODULES

This section presents the results of the replacement analysis applied to the self-attention modules of Gemma 2 (2B).

Figure 15 highlights the performance difference on HellaSwag between Gemma 2 (2B) with self-attention layers replaced by those from another block and the model with these layers entirely removed. The heatmap reveals a heterogeneous pattern, where most cells indicate that the replacement of the self-attention results in performance degradation comparable to or worse than its ablation. In a few cases, replacement provides slight improvements, primarily in substitutions of the layers within block 1 and in specific configurations of replacements among central layers. A similar trend is observed in Figure 18, where replacement generally leads to worse performance than removal, with a few exceptions in mid-block self-attention modules.

The distribution of red cells shows partial consistency in the middle layers across plots, but still functional redundancy appears to be weakly represented in the model.

Ultimately, the replacement analysis indicates that using a self-attention transformation from one block to approximate another is ineffective, reinforcing the notion of minimal inter-sub-block redundancy in Gemma 2 (2B), as previously suggested by the analysis on FFNN layers.

### A.2 COMPARISON BETWEEN LLAMA 3.1 AND GEMMA 2 (2B)

The comparison of experimental results between Llama 3.1 and Gemma 2 (2B) reveals significant differences between the two models. Performance degradation trends are inconsistent across them, with the most critical blocks varying between architectures. Additionally, similarity patterns differ significantly between the two models, suggesting that redundancy, if present, is highly dependent on the specific model architecture.

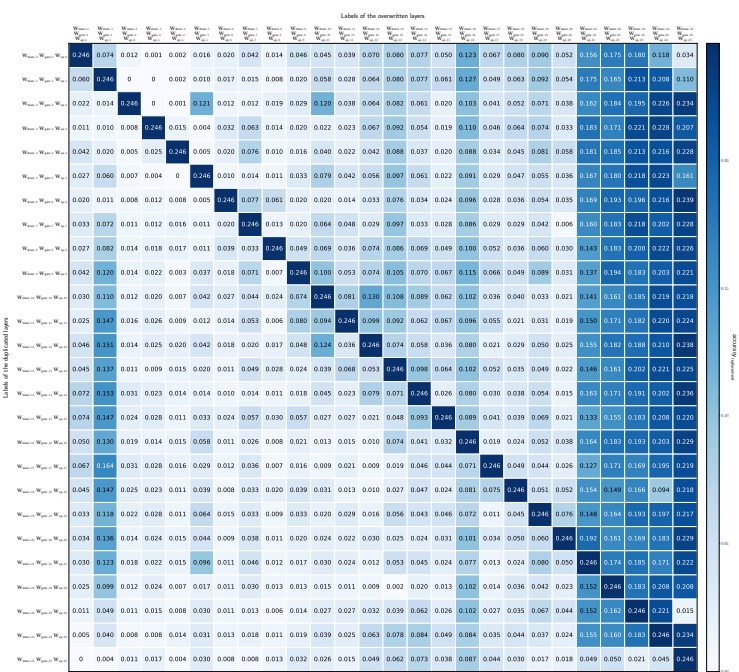

Figure 10: Heatmap illustrating the accuracy on GSM8K of Gemma 2 (2B) having linear layers of the FFNN module replaced by the ones of a different block.

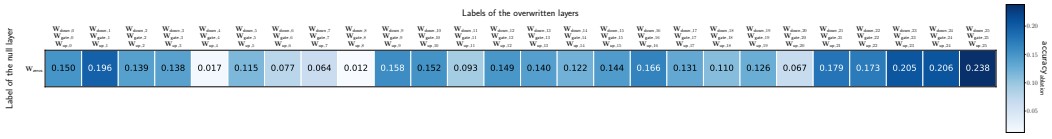

Figure 11: Heatmap illustrating the accuracy on GSM8K of Gemma 2 (2B) having linear layers of the FFNN module removed.

Gemma 2 (2B) appears to have functional modules that behave more distinctly from one another compared to Llama 3.1. Heatmaps illustrating performance deltas between LLM with replaced layers and those with ablated layers show a broader and stronger presence of negative values in Gemma 2. In both self-attention and FFNN modules, substituting a module with one from a different block tends to degrade performance more severely than in Llama 3.1, suggesting lower functional redundancy. The reduced redundancy could be attributed to the smaller size of Gemma 2 (2B), which may result in lower over-parameterization and fewer repeated computations across layers.

The inconsistencies observed comparing Gemma 2 and Llama 3.1 highlight significant differences in how concepts are learned and represented across layers in these Transformer-based architectures. The two models differ in terms of knowledge localization and distribution, emphasizing how variations in training processes, architectural modifications, and dataset composition lead to fundamentally different "reasoning patterns" in the resulting models.

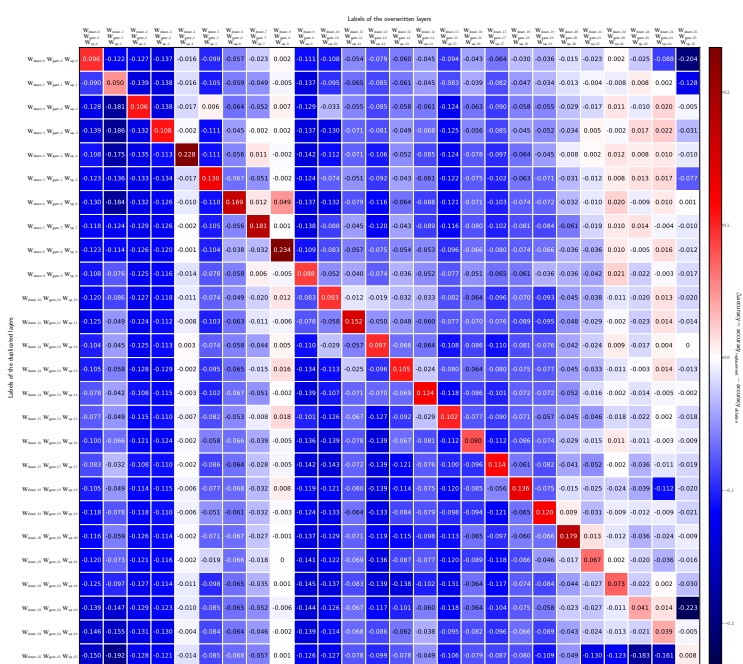

Figure 12: Heatmap illustrating the difference in performance evaluated on GSM8K between Gemma 2 (2B) with linear layers in the FFNN module replaced by those from another block and Gemma 2 (2B) with the same layers removed.

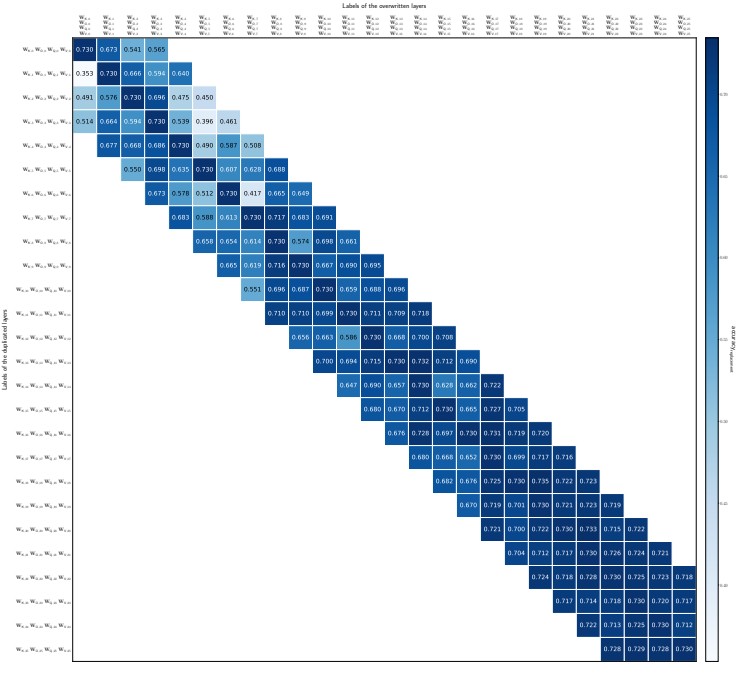

Figure 13: Heatmap illustrating the accuracy on HellaSwag of Gemma 2 (2B) having linear layers of the self-attention module replaced by the ones of a different block.

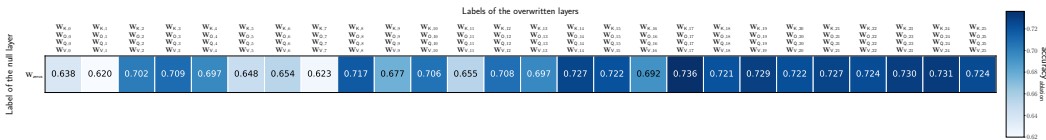

Figure 14: Heatmap illustrating the accuracy on HellaSwag of Gemma 2 (2B) having linear layers of the self-attention module removed.

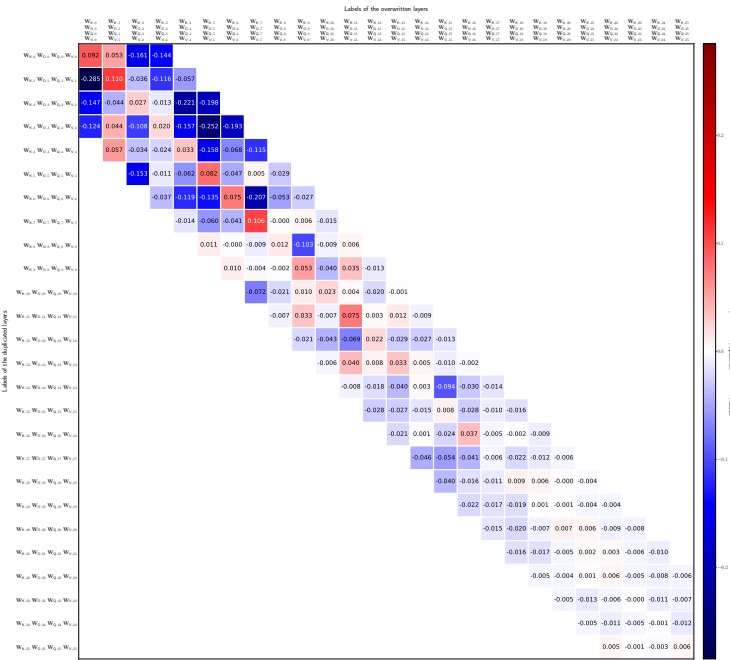

Figure 15: Heatmap illustrating the difference in performance evaluated on HellaSwag between Gemma 2 (2B) with linear layers in the self-attention module replaced by those from another block and Gemma 2 (2B) with the same layers removed.

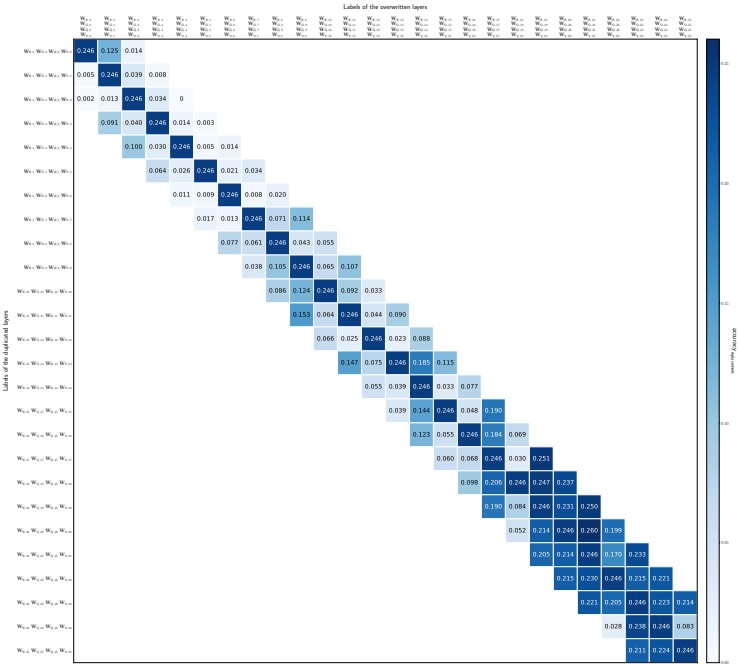

Figure 16: Heatmap illustrating the accuracy on GSM8K of Gemma 2 (2B) having linear layers of the self-attention module replaced by the ones of a different block.

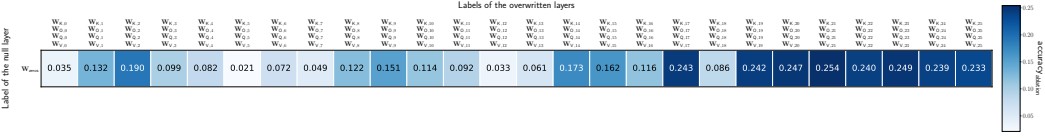

Figure 17: Heatmap illustrating the accuracy on GSM8K of Gemma 2 (2B) having linear layers of the self-attention module removed.

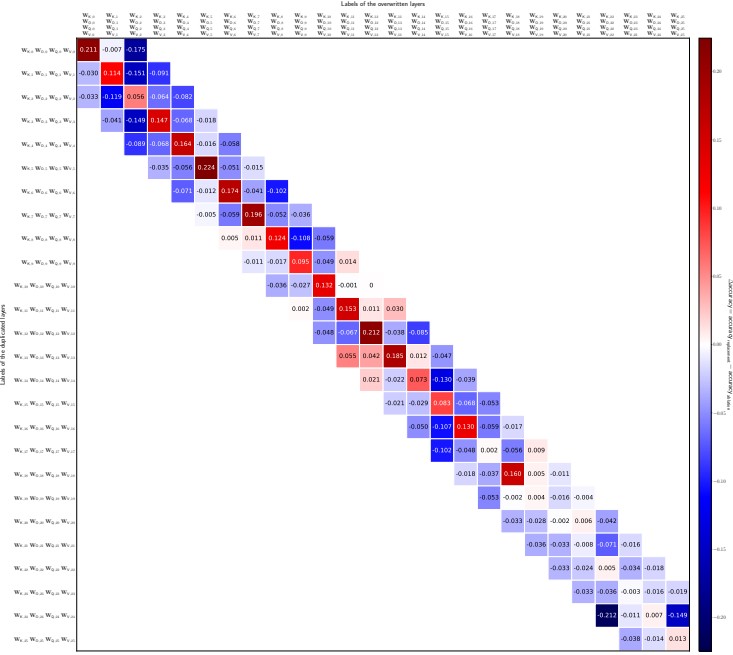

Figure 18: Heatmap illustrating the difference in performance evaluated on GSM8K between Gemma 2 (2B) with linear layers in the self-attention module replaced by those from another block and Gemma 2 (2B) with the same layers removed.