# OpenReview forum: "Exploring Redundancy and Shared Representations for Transformer Models Optimization"
_ICLR.cc/2026/Conference — Submitted to ICLR 2026_

### Official Review · Reviewer_vBWj · 2025-10-26

**Soundness:** 2
**Presentation:** 2
**Contribution:** 2
**Rating:** 2
**Confidence:** 4

**Summary:**

The paper studies functional redundancy across Transformer blocks and proposes three compression strategies designed to exploit potential sharing: MASS groups same-shape matrices by type and replaces them with a simple average; GlobaL Fact introduces a shared global low-rank factor per group and per-layer local factors; ABACO couples low-rank adapters with an exponentially decaying contribution from the original weights during fine-tuning. The central empirical probe of redundancy swaps modules (primarily FFNNs) between blocks and measures downstream performance. Experiments show that direct sharing or factorization causes substantial accuracy drops and that inter-block redundancy is weaker than expected.

**Strengths:**

1. The paper takes a principled look at whether different blocks perform overlapping functions and adopts a concrete,  performance-based replacement test to operationalize "functional redundancy." The results verified some results that were in line with the intuitions of model training (such as the importance of the first and last layers of the model). This provides insght for the design of model compression algorithms.
2. The three proposed strategies are straightforward to implement, which lowers the barrier for follow-up work. The paper openly reports when methods collapse and does not overstate claims.

**Weaknesses:**

1. The analysis method of pairwise substitution one by one in Section 3 has an extremely high computational complexity ($N^2$) and is difficult to continue to be used on larger models. At the same time, the selection of datasets and tasks leads to strong task-specific conclusions (for example, outside the early layers of HellaSwag, attention is not so important), which may not hold true elsewhere.
2. Limited novelty of the proposed methods: MASS directly reduces to uniform averaging within groups of weight matrices without any alignment or importance weighting, so it is unsurprising that capacity is lost. GlobaL Fact is essentially shared low-rank factorization initialized from group SVD, a close cousin of standard truncated-SVD/LoRA-style factorization, with modest adjustments (shared/global vs local). Conceptual distance from known techniques feels small. ABACO is LoRA with an annealed scalar $\alpha$ on the original weights. While neat in practice, the mechanism is incremental rather than a new compression principle. Most importantly, these methods do not truly address the issue identified in Section 3 of the article (redundancy in the model), which is to remove redundancy while ensuring model performance.
3. The author only conducted tests on two benchmarks (HellaSwag and GSM8K) and models (Llama-3.1-8B and Gemma-2-2B), and the reliability and comprehensiveness of the experimental results were insufficient. No further verification was conducted on models such as Qwen3 and DeepSeeking R1 for tasks like language modeling confusion, instruction compliance, or reasoning evaluation.
4. The paper does not provide a systematic report on the actual speed, memory usage, or throughput/latency changes of the proposed method, which is crucial for the model compression.

**Questions:**

Please refer to the "Weakness" section.

---

> ### Author Response · Authors · 2025-12-01
>
> We are grateful for the reviewer’s comments, and are working on improving our work based on the suggestions received, particularly the suggestions regarding speed and memory usage.

---

### Official Review · Reviewer_DuWY · 2025-10-27

**Soundness:** 2
**Presentation:** 2
**Contribution:** 1
**Rating:** 2
**Confidence:** 2

**Summary:**

This paper focuses on model compression for LLMs. Authors argue that the FFNN and attention module have pretty big redundancy and propose to ground the parameters that are close enough and use the average of it to merge into a shared layer using the mean aggregation of different layers. In addition, authors have conducted further experiemnts with local and global factorization as well as adapter-based approximation. Authors have conducted experiments on HellaSwag and GSM8K, where the results on the HellaSwag shows comparable results after finetuning.

**Strengths:**

+ The overall experiment is comprehensive and show reasonable analysis.

**Weaknesses:**

- The overall compression is showing serious degradation for the three introduced method for the final performance matrix, and in some case it is showing purely random results. It is not likely to be easily applied on other models for better generalizations and the downstream use case is limited.

- This paper is working as a paper to have negative results, while the exploration of the model and dataset selection is quite limited. As a paper with negative results, it is strongly suggested to have some wider selection of model and dataset, as well as multiple metrics, for further in-depth analysis.

- Please consider change the presentation in the experiment part, as showing your results of aggregating all your models, and then showing different variations as ablations.

**Questions:**

N/A

---

> ### Author Response · Authors · 2025-12-01
>
> We appreciate the reviewer’s insightful comments. We agree that the paper requires further refinement and are already working to address the issues highlighted, specifically regarding the limited scope of the contribution.

---

### Official Review · Reviewer_NX5E · 2025-10-29

**Soundness:** 1
**Presentation:** 1
**Contribution:** 1
**Rating:** 0
**Confidence:** 5

**Summary:**

In my opinion, this paper was written by an LLM without any review or editing.
The most critical issues are as follows.
1. Significant quality issue (lots of typos in all tables, and extremely small fonts in all figures).
2. Lack of experimental results (only HellaSwag and GSM8K).
3. Significant accuracy degradation after compression (Accuracy 0~0.01 on GSM8k).

Thus, I recommend "Strong Reject".

**Strengths:**

.

**Weaknesses:**

See summary

**Questions:**

.

**Details Of Ethics Concerns:**

By submitting a paper drafted exclusively by an LLM without human oversight, the authors are wasting reviewers' time and potentially sharing misleading information through the OpenReview system.

---

> ### Author Response · Authors · 2025-12-01
>
> We would like to clarify that this manuscript is the result of original research and that the writing is our own. Language models were ONLY used for minor editorial polishing, as disclosed in the submission. While we thank reviewer NX5E for their time and insights regarding our paper, the allegation that the paper was somehow “drafted exclusively by an LLM without human oversight” is completely wrong and inappropriate given the amount of effort we went to for its preparation.
>
> Regarding the specific critiques:
>
> 1. Typos and errors: We have carefully re-examined the tables and noted no formatting errors. We note that the zero accuracy values in some tables are not typos, but the negative outcomes of the compression method applied to reasoning tasks. However, we understand they can be misinterpreted as typos or errors, and will therefore remove them from future version of the paper for more clarity.
> 2. Scope: We acknowledge that the current experimental scope is limited and that the negative findings require more comprehensive benchmarking to strengthen the conclusions.

---

### Official Review · Reviewer_Vn3A · 2025-11-01

**Soundness:** 2
**Presentation:** 2
**Contribution:** 1
**Rating:** 2
**Confidence:** 4

**Summary:**

This paper investigates functional redundancy in Transformer-based language models by systematically replacing modules (FFNNs and self-attention) across layers and measuring performance impact on HellaSwag and GSM8K benchmarks.

The results show that redundancy in Transformer models is lower than previously assumed. When replacing layers, performance typically degrades more than when ablating layers. Based on this finding, the authors propose three compression methods: MASS (Matrix Aggregation and Sharing Strategy), GlobaLFact (Global and Local Factorization), and ABACO (Adapter-Based Approximation and Compression Optimization).

Experimental results on Llama 3.1 and Gemma 2 show that all three methods have limited success, with significant performance degradation even after fine-tuning. The authors conclude that Transformers exhibit less exploitable redundancy than expected.

**Strengths:**

The author addresses an important and timely issue of LLM efficiency and compression. They provide transparent reporting of negative results, which is valuable for the research community. The authors acknowledge the limitations of their approach and avoid overstating their results.

**Weaknesses:**

Limited novel contribution -- main finding is negative results without sufficient insights into why redundancy is hard to exploit. Experimental scope is limited: only 2 models, 2 benchmarks, no statistical significance testing or multiple runs.
Replacement analysis methodology is relatively crude – swapping entire modules doesn't capture nuanced forms of redundancy or partial similarity.

Missing comparisons with compression baselines (especially pruning and factorization methods).
The paper doesn't explore more sophisticated grouping criteria or aggregation strategies despite acknowledging their limitations.
Authors cite sophisticated activation-aware factorization methods in Related Work but then use naive SVD in their own approach. This makes the comparison unfair and the negative results less meaningful -- of course basic SVD fails when papers like FWSVD, ASVD and SVD-LLM have shown you need activation (or gradient)-aware methods.

The few cases of successful replacement (positive redundancy) deserve deeper analysis. The title promises exploration of redundancy, but the paper is really about "why (naive) compression methods fail"

**Questions:**

If the paper is about the level of redundancy being lower than expected, could you please clearly state what the previous belief in the field was?
Why were these datasets chosen? Based on previous research [1][2], there are assumptions about different levels of quality decline for tasks with varying levels of complexity.

Heat maps (Figures 1-4 and 7-18) are challenging to read and understand, as they lack clear visual patterns in dense numerical matrices. Could you please consider reformatting them or using other visualization techniques?
Do you have evidence that such a rough combination of modules into an FFNN module is generally compressible? Because there are articles that explicitly prove the importance of not just individual modules, but individual neurons [3].
Why not compare against modern compression baselines like quantization (GPTQ, AWQ), recent pruning methods beyond SparseGPT, or modern factorization methods?


[1] Yin, L., Jaiswal, A., Liu, S., Kundu, S., & Wang, Z. (2023). Pruning small pre-trained weights irreversibly and monotonically impairs" difficult" downstream tasks in llms. arXiv preprint arXiv:2310.02277.

[2] Frankle, J., & Carbin, M. (2018). The lottery ticket hypothesis: Finding sparse, trainable neural networks. arXiv preprint arXiv:1803.03635.

[3] Yu, Mengxia, et al. "The super weight in large language models." arXiv preprint arXiv:2411.07191 (2024).

---

> ### Author Response · Authors · 2025-12-01
>
> We thank the reviewer for the insightful comments. We agree the work should explore more sophisticated compression strategies, and are working on expanding the methodologies tested in that direction.

---

### Meta-Review · Area_Chair_omtC · 2026-01-12

**Summary:**

All four reviewers recommend rejection. Reviewers acknowledge that understanding redundancy and compression in large language models is important and appreciate the transparent reporting of negative results. However, the paper’s scientific contribution is weak: the main conclusion is largely negative without delivering new conceptual insight into why redundancy is difficult to exploit. The experimental scope is narrow, the redundancy analysis is coarse (entire-module replacement), and the proposed compression methods are viewed as naive or incremental relative to existing state-of-the-art baselines. Significant performance degradation after compression, lack of comparisons with modern pruning, quantization, or activation-aware factorization methods, limited analysis of positive cases, and missing system-level metrics (speed/memory) collectively undermine the paper’s impact and credibility.

**Reviewer Concerns:**

The authors appropriately acknowledge several key limitations raised by reviewers, including the narrow experimental scope, the need for more sophisticated compression strategies, and the fact that negative results require broader validation to be convincing. They also clarified that zero or near-zero accuracy values reflect genuine failures of the compression methods rather than typographical errors, and firmly addressed the allegation that the paper was written entirely by an LLM, stating that language models were used only for minor editorial polishing. These responses help resolve misunderstandings around presentation quality and authorship intent, and demonstrate awareness of the paper’s current shortcomings.

The core technical criticisms remain unresolved. The paper still lacks comparisons with modern, activation-aware compression and factorization baselines, making the negative results difficult to interpret or contextualize. The redundancy analysis remains overly coarse and computationally expensive, without probing finer-grained structure (e.g., neuron- or activation-level redundancy). The experimental evaluation is still limited to two models and two benchmarks, with no statistical robustness, no diverse task coverage, and no reporting of practical compression benefits such as speedup or memory savings. Several reviewers also note that the proposed methods do not meaningfully address the redundancy findings they are motivated by, and that the paper reads more as evidence that naive compression fails than as a principled study of redundancy in Transformers.

**Reviewer Scores:**

The authors clarified only authorship and formatting issues; consequently, none of the reviewers’ scores were updated.

Reviewer Vn3A: 2.

Reviewer NX5E:0.

Reviewer DuWY: 2.

Reviewer vBWj: 2.

---

### Decision · Program_Chairs · 2026-01-26

Reject